# Exploring the Potential Use of Natural Products Together with Alkalization in Cancer Therapy

**DOI:** 10.3390/pharmaceutics16060787

**Published:** 2024-06-10

**Authors:** Masahide Isowa, Reo Hamaguchi, Ryoko Narui, Hiromasa Morikawa, Toshihiro Okamoto, Hiromi Wada

**Affiliations:** 1Japanese Society on Inflammation and Metabolism in Cancer, 119 Nishioshikouji-cho, Nakagyo-ku, Kyoto 604-0842, Japan; misowa@scim.or.jp (M.I.); rnarui@scim.or.jp (R.N.); hmorikawa@scim.or.jp (H.M.); wadah@kuhp.kyoto-u.ac.jp (H.W.); 2Department of Thoracic and Cardiovascular Surgery, Cleveland Clinic, Cleveland, OH 44195, USA; okamott@ccf.org

**Keywords:** tumor microenvironment, alkalization therapy, natural products, cancer, complementary and alternative medicine

## Abstract

Cancer treatment is a significant focus in medicine, owing to the increasing global incidence of cancers. Patients with advanced cancers that do not respond to conventional therapies have limited options and an unfavorable prognosis. Consequently, researchers are investigating complementary approaches to conventional treatments. One such approach is alkalization therapy, which aims to neutralize the acidic tumor microenvironment (TME) by increasing its pH level. The acidic TME promotes inflammation, tumor progression, and drug resistance. Alkalization therapy has been demonstrated to be effective for various cancers. In addition, natural products, such as triterpenoids, parthenolides, fulvic acid, *Taxus yunnanensis*, and apple pectin have the potential to alleviate symptoms, maintain physical fitness, and improve treatment outcomes of cancer patients through their anti-inflammatory, antioxidant, and anticancer properties. In this review, we focus on the effects of alkalization therapy and natural products on cancer. Furthermore, we present a case series of advanced cancer patients who received alkalization therapy and natural products alongside standard treatments, resulting in long-term survival. We posit that alkalization therapy together with supplementation with natural products may confer benefits to cancer patients, by mitigating the side effects of chemotherapy and complementing standard treatments. However, further research is warranted to validate these clinical findings.

## 1. Introduction

Cancer is a serious public health problem that is increasing worldwide, and its treatment has become an important focus of medicine. Despite the recent development of innovative cancer therapies, and improvements in the survival and quality of life of cancer patients, the management of patients with advanced cancer remains a difficult challenge [1]. In particular, patients with advanced cancers that do not respond adequately to existing therapies, such as surgery, radiation therapy, and chemotherapy have limited treatment options, and often have an unfavorable prognosis [2]. In addition, these standard therapies are double-edged swords, sometimes resulting in a variety of health problems owing to adverse events. To overcome these difficulties, researchers and clinicians are investigating complementary approaches to conventional therapies.

It is well known that energy metabolism in malignant tumors differs from that in normal cells and is dependent on the glycolytic system. This concept was proposed by Otto Warburg and colleagues and is known as the Warburg effect [3]. The specific mechanism of tumor energy metabolism and its effects on the tumor microenvironment (TME) have been the focus of much attention and research to date [4]. The TME is a dynamic and complex entity that substantially affects cancer development and progression. It is not just a collection of tumor cells, but includes immune cells, stromal cells, blood vessels, and the extracellular matrix, which interact with tumor cells, affecting tumor initiation, progression, and therapeutic responses [5,6]. Malignant tumors produce high levels of lactate, or H^+^, as a result of enhanced aerobic glycolysis, and the extracellular efflux of H^+^ leads to acidification of the TME [7]. The intracellular and extracellular pH of both normal and tumor cells have been measured in various tissues, and in general, the extracellular pH of tumor cells is 0.3 to 0.7 pH units lower than the average extracellular pH of normal tissues [8]. An acidic TME attracts inflammatory cells, leading to chronic inflammation, tumor progression, and drug resistance [3]. In our clinic, we have implemented “alkalization therapy”, aimed at neutralizing the acidic TME, and have achieved some efficacy in various cancers [9,10,11,12].

Furthermore, in addition to standard cancer treatments, many patients are now trying natural products (i.e., antioxidants, vitamins, herbs, and other natural remedies) that have been reported to reduce symptoms, maintain physical fitness, and possibly improve treatment outcomes [13]. Various natural products have been shown to inhibit the growth, migration, and invasion of cancer cells owing to their anti-inflammatory, antioxidant, and even anticancer properties [14]. Plants and foods containing these compounds have long been used in traditional medicine in many parts of the world and are now available as medical supplements and dietary/health food items, and are expected to have various effects. However, the clinical efficacy and function of these natural products remain to be elucidated.

In this review, we will provide an overview of our approach to alkalization of the TME and the effects of various natural products on cancer, including a literature review. We also report a cohort of patients from our clinic with advanced cancer who had metastasis and postoperative recurrence, in whom long-term survival was achieved by alkalization therapy and treatment with natural products, in addition to standard treatments.

## 2. “Alkalization Therapy” for Cancer

### 2.1. Concept of Alkalization Therapy

“Alkalization therapy” is simply defined as “systemic alkalization of the body using diet and alkalizing agents to increase the pH of the local TME of the tumor”. The theory of this therapy is as follows. First, tumor cells in malignant tumors have damaged mitochondria, and consequently use the glycolytic system for cellular metabolism (the Warburg effect) [15]. This results in malignant tumor cells having 10- to 40-fold higher rates of glucose uptake than normal cells, leading to 10- to 100-fold higher rates of lactate production, and subsequently causing lactate accumulation and acidification outside of the tumor cells [15]. This acidic environment, or decreased TME pH [7], attracts inflammatory cells, leading to chronic inflammation [3], and is also associated with distant metastasis [16], creating an environment that is more favorable for cancer survival. Various other molecular mechanisms have also been focused on as the cause of pH dysregulation in the peritumoral environment [17,18]. Alkalization therapy may have the potential to regulate the TME and has hence attracted attention as a possible novel anticancer therapy target [8,19].

### 2.2. Studies Demonstrating the Positive Effects of Alkalization Therapy on Cancer

Hirschhaeuser et al. have shown from various basic research studies that lactic acid produced from tumors is involved in the early growth process of cancer and in blunting the tumor immune response [20]. In hepatocellular carcinoma (HCC) cells, pH-regulating molecules, such as carbonic anhydrase IX and XII, and vacuolar-type ATPase act as a bridge between tumor cells and their TME, which may help to inhibit tumor progression and reduce immunosuppression [21]. Several studies have reported that increasing the pH of the acidic TME can have anticancer effects. Pötzl et al. reported that systemic alkalization by the oral administration of bicarbonate enhances natural killer cell interferon-gamma expression, and significantly slows lymphoma growth in mice [22]. The oral administration of bicarbonate to mice reduced breast cancer invasion [23], and inhibited distant metastasis in mouse models of breast and prostate cancer [24]. The oral administration of alkaline water (an alkalizing agent dissolved in water) to mice substantially reduced the growth of malignant melanoma [25], and a combination of high-dose proton pump inhibitors and alkaline water together with chemotherapy increased the anticancer effects and extended the life span of dogs and cats with advanced cancer [26]. In a transgenic mouse model of adenocarcinoma of the prostate, oral administration of alkaline water or NaHCO_3_ reduced tumor growth and the progression from high-grade prostatic intraepithelial neoplasia (PIN) to prostate adenocarcinoma [27]. This result demonstrated that alkalization therapy prolongs the time that tumors retain a more differentiated low-grade PIN state, suggesting that alkalizing agents work not only as tumor suppressants, but also as tumor-preventing agents.

Furthermore, some reports have suggested that alkalizing agents, such as bicarbonate, may not only inhibit local tumor invasion and growth by alkalizing the TME, but may also have a synergistic effect with chemotherapy and immunotherapy [28,29,30,31,32]. In vitro, a low pH reduces the uptake and cytotoxicity of weakly basic chemotherapeutic drugs, such as anthracyclines (doxorubicin, daunorubicin, and mitoxantrone, etc.), and this phenomenon has been thought to contribute to the physiological resistance to these drugs in vivo [33]. A study demonstrated that doxorubicin has higher toxicity at a higher pH in Michigan Cancer Foundation-7 human breast cancer cells and in severe combined immunodeficiency mice [30]. Theoretical calculations of mitoxantrone uptake into cells indicated that an increase in the therapeutic index of up to 3.3 fold is possible with NaHCO_3_ pretreatment of C3H tumor-bearing C3H/HeN mice [32].

Regarding the association between diet and cancer, diets rich in fruits and vegetables may be protective against cancer, owing to bioactive components, such as organosulfur compounds, berry compounds (vitamins and flavonoids contained in berries), and cruciferous vegetable-derived isothiocyanates [34]. Studies have shown that specific foods have different systemic buffering effects [35]. In addition, Robey et al. have shown that animal protein and salt make the body more acidic, whereas fruits and vegetables make the body more alkaline, suggesting that changes in body pH caused by diet may affect molecular mechanisms at the cellular level [36]

There are still only a limited number of studies that have applied the idea of alkalization of the TME as an anticancer treatment in clinical practice. Park et al. estimated the dietary acid load of patients using potential renal acid load (PRAL), and concluded that breast cancer incidence was highest in the group with the highest PRAL, in which patients had a diet that was high in meat consumption and low in fruit and vegetable intake, and hence a diet with limited meat and high fruit and vegetable intake may reduce breast cancer incidence [37]. Some studies focusing on clinical symptoms and quality of life rather than antitumor effects have shown that the infusion of dimethyl sulfoxide and sodium bicarbonate significantly improved clinical symptoms and quality of life in metastatic prostate cancer [38] patients, whereas another study reported a reduction in pain by systemic body alkalization using intravenous dimethyl sulfoxide and sodium bicarbonate infusion in difficult-to-treat terminal cancer patients [39].

Mathematical model simulations as well as experimental observations have demonstrated that bicarbonate effectively increases the extra-tumoral pH (extra-tumoral pH was predicted to increase from 7.0 to 7.07 in mice; and was predicted to increase from 7.0 to 7.04 in humans), and that chronic oral bicarbonate administration was effective enough to be a potential anticancer treatment [40]. Furthermore, a phase 0/1 clinical trial using sodium bicarbonate for cancer treatment proposed a specific prescription regimen that is feasible and safe [41].

A concept that has recently been proposed as a target for anti-tumor therapy is “alkaliptosis” [42]. This is a new concept in which regulated cell death is promoted by increasing the intracellular pH. Although this is different from our original goal of alkalization therapy, which is to correct the pH dysregulation of tumors by increasing the pH of the TME (which is extracellular) to obtain an anticancer effect, alkaliptosis may also result in an anticancer effect. This concept hence indicates that the effects of alkalization therapy should be investigated further using theoretical insights.

Research collectively suggests that systemic alkalization may not only have antitumor effects and alleviate symptoms but may also prevent tumor formation. It holds promise for boosting chemotherapy efficacy and stimulating tumor immunity. Therefore, alkalization therapy, which modifies the host’s environment, offers a strategic approach to make conditions less favorable for tumor survival, and complements other treatments, such as natural products.

### 2.3. Definition of Alkalization Therapy

We define alkalization therapy as a combination of an alkalizing diet and oral intake of sodium bicarbonate and/or sodium potassium citrate as an alkalizing agent [11]. An “alkalizing diet” is defined as a diet containing a large number of fruits and vegetables, with blue-back fish as the main source of protein, and as little meat and dairy as possible [43,44]. Specifically, patients are instructed to consume at least 400 g of fruits and vegetables daily, and to keep a dietary record for the first 4 weeks. In addition, when patients visit our clinic, the doctor and nurses review their dietary records to ensure that their diet meets our criteria of an alkalizing diet and advise patients on dietary adjustments that need to be made. Ultimately, the choice of diet is left to the patient. We believe that any current chemotherapies or the additional intake of natural products may have minimal effects if a proper alkaline diet is not followed, and hence this is the treatment we focus on the most.

### 2.4. Alkalization Therapy; Our Clinical Experience to Date

In our clinic, we have empirically combined alkalization therapy with chemotherapy, and found that in some patients with various solid tumors and hematologic tumors, alkalization therapy shows substantial therapeutic effects [11]. In a comparative study of advanced pancreatic cancer patients, either treated with conventional chemotherapy plus alkalization therapy or with chemotherapy alone, the median overall survival (OS) was significantly longer (15.4 months vs. 10.8 months; *p* < 0.005) and the mean urine pH was significantly higher in the chemotherapy plus alkalization therapy group than in the chemotherapy alone group (6.80 ± 0.71 vs. 6.38 ± 0.85; *p* < 0.05). This suggested that alkalization therapy contributed to the increase in urine pH, which we consider to be a marker of whole-body alkalization, by acting as a systemic buffering therapy, and had a synergistic effect with chemotherapy, thus leading to the prolongation of patient OS [10]. In our study focusing on stage 4 pancreatic cancer patients treated with alkalization therapy combined with conventional therapy [45], we divided patients into three groups by their mean urine pH. In the 98 patients analyzed, the median OS from the time of diagnosis was 13.2 months (95% confidence interval [CI] = 9.7–16.1 months). Patients with a mean urine pH of 7.5 or greater had a median OS of 29.9 months (95% CI = 9.1–38.7) compared with 15.2 months (95% CI 10.1–21.2) for those with a mean urine pH of 6.5 to 7.5, and 8.0 months (95% CI = 5.6–15.5) for those with a mean urine pH of less than 6.5. This demonstrated a trend of a longer OS in patients with a higher urine pH (*p* = 0.039), in whom alkalization therapy was thought to be successful. In a study on patients with HCC, patients treated with alkalization therapy were divided into two groups, namely, a group with a mean post-treatment urine pH of ≥7.0, and the other with a pH of <7.0. The results showed that the median OS from the start of alkalization therapy of patients with a urine pH of ≥7.0 was not reached (*n* = 12, 95% CI = 3.0—not reached), which was significantly longer than that of patients with a pH of <7.0 (15.4 months, *n* = 17, 95% CI = 5.8—not reached, *p* < 0.05) [12]. Furthermore, in an observational study of small cell lung cancer patients, alkalization therapy and intravenous vitamin C combined with chemotherapy (intervention group) was compared with chemotherapy alone (control group), and the mean urine pH of the intervention group was found to be significantly higher than that of the control group (7.32 ± 0.45 vs. 6.44 ± 0.74; *p* < 0.05). The median OS of the intervention group was 44.2 months (95% CI = 22.0—not reached), compared with 17.7 months for the control group (95% CI = 13.5—not reached; *p* < 0.05) [9]. In terms of combination treatment using alkalization therapy and tyrosine kinase inhibitors (TKIs), we retrospectively evaluated 11 advanced or recurrent non-small cell lung carcinoma patients with epidermal growth factor receptor (EGFR) mutations, who were treated with EGFR-TKI after being instructed to follow an alkaline diet [44]. The median progression-free survival and OS were 19.5 (range = 3.1–33.8) and 28.5 (range = 15.4–46.6) months, respectively. The average dosage of EGFR-TKI was 56% ± 22% of the standard dosage used for standard chemotherapy. Urine pH was significantly increased after starting an alkaline diet (6.00 ± 0.38 [before] vs. 6.95 ± 0.55 [after]; *p* < 0.05). We also reported a successful treatment response to the combination of alkalization therapy and/or intravenous vitamin C infusion and/or natural products administration in patients with unresectable renal pelvis cancer, malignant lymphoma of the stomach, malignant lymphoma of the tonsils, unresectable gastric adenocarcinoma with multiple liver metastases, multiple gastric cancer, postoperative recurrence of gastric cancer, and postoperative recurrence of breast cancer with multiple systemic metastases, including in the bones and lungs. These results and cases suggest that alkalization therapy may have a therapeutic effect on its own, and a synergistic effect with other anticancer therapies are also expected.

## 3. Natural Products for the Treatment of Cancer

### 3.1. Triterpenoids

Triterpenoids (Figure 1) constitute compounds within the terpenoids, which is the most extensive and diverse class of natural substances [46]. They encompass a broad spectrum of structural variations, from acyclic to hexacyclic C-30 skeletons, reflecting the extensive diversity and complexity inherent in natural compounds [46]. The triterpenoid compounds ursolic acid and oleanolic acid are isomers, and are widely found in foods, medicinal herbs, and other plants [47]. Ursolic acid and oleanolic acid tend to be discussed as a pair, but here we would like to focus on ursolic acid, which can be extracted from Japanese plums, and have long been used as a health food in Japan. Ursolic acid, which is also known as 3β-3-hydroxy-urs-12-ene-28-oic-acid, is a pentacyclic triterpenoid, which has the chemical formula C_30_H_48_O_3_ and a molecular mass of 456.71 g/mol [48]. It has been used in folk medicine in China and in Ayurvedic medicine, and is also found in apples, basil, bilberries, cranberries, peppermint, rosemary, and oregano, etc. Pharmacologically, ursolic acid is known for its hepatoprotective, anti-inflammatory, antihyperlipidemic, and anticancer effects [47]. In terms of anticancer effects, its ability to affect the activity of several intracellular enzymes enables it to modulate processes that occur within tumor cells and activate pathways leading to apoptosis (programmed cell death), including inhibition of the MAPK/ERK and PI3K/ACT/mTOR signaling cascades [49]. This activation of pathways that lead to apoptosis is the most important function of ursolic acid’s anticancer activity, resulting in the inhibition of pathways that lead to cancer proliferation, growth, and metastasis. Other anticancer activities of ursolic acid have also been reported, such as its effects on changes that occur upon the exposure of cells to carcinogenic chemicals (e.g., benzo(a)pyrene [50] and substances extracted from tobacco smoke [51]), reactive oxygen species (ROS) [52], ionizing radiation [53], and Epstein–Barr virus [54].

### 3.2. Parthenolide

Parthenolide (Figure 2) is a type of sesquiterpenoid derived from the leaves and flowers of feverfew (*Tanacetum parthenium*) and is a natural compound that has been used as an herbal remedy since ancient times [55]. It has the chemical structure C_15_H_20_O_3_ and a molecular mass of 248.32 g/mol [56], and causes anti-inflammatory and anticancer effects by strongly inhibiting the activity of nuclear factor kappa-B (NF-κB) [57]. NF-κB is an important transcription factor involved in cancer cell proliferation, survival, invasion, metastasis, and evasion of antitumor immunity. To date, parthenolide has been shown to have high cytotoxicity and apoptosis-inducing effects against malignant tumors, such as breast cancer and chronic myeloid leukemia, and to inhibit the proliferation and self-renewal ability of cancer stem cells, suggesting its potential usefulness for the curative treatment of cancer [58,59]. In addition, a study has shown that parthenolide has anticancer effects by targeting EGFR in non-small cell lung cancer [60]. Furthermore, parthenolide inhibits the mitochondrial respiratory chain and increases the production of ROS [61]. ROS cause DNA damage and apoptosis in cancer cells, and hence parthenolide can also be expected to have anticancer effects by reducing the expression of antioxidant enzymes and increasing the sensitivity of cancer cells to oxidative stress.

### 3.3. Fulvic Acid

Fulvic acid (C_14_H_12_O_8_, molecular mass: 308.24 g/mol [62], Figure 3) is a high-molecular weight organic acid resulting from the decomposition of ancient plants and animals by microorganisms [63]. Fulvic acid is also the main component of Shilajit, an organic mineral product that has been applied in the field of Ayurvedic traditional medicine for a long time [64]. The actions of fulvic acid in the body have been summarized in previous reports as immunomodulatory, oxidation-regulating, and gastrointestinal-activating, and it has been found to promote the activation of various physiological functions [65]. Regarding anticancer effects, fulvic acid is considered to prevent the progression of cancer by inhibiting the proliferation of cancer cells and inducing apoptosis [66,67]. In addition, a study has suggested that the binding of fulvic acid to transferrin in human serum enables more efficient delivery of antitumor drugs to the target tumor [68].

### 3.4. Taxus yunnanensis (Taxus Plant)

Paclitaxel, extracted from a plant called *Taxus brevofolia*, was originally known as the anticancer drug taxol. It is widely used in large quantities in both clinical and basic research and is well-known as a clinically effective anticancer molecule [69,70]. One of its sources, *T. yunnanensis*, is in the same genus as *T. brevofolia*, and is endemic to China, but is an endangered species that is being depleted as a biological resource [71]. Unlike other species of the *Taxus* genus, *T. yunnanensis* has a high taxol concentration [72], and has unique polysaccharides that have been shown to substantially inhibit the proliferation of HeLa and HT1080 cells in a concentration-dependent manner [73]. *T. yunnanensis* has a special position within the genus, as it contains a large amount of α-conidendrin, which induces apoptosis in breast cancer cell lines [74]. There are also reports that *T. yunnanensis* has significantly higher oral absorption and bioavailability than pure extracted paclitaxel in rat studies [75], and it has been associated with the induction of tumor cell apoptosis via multiple pathways [76].

### 3.5. Apple Pectin

Pectin is a substance obtained from the remains of the extraction of sugar and juice from fruits, and it plays an important role as a soluble plant fiber. However, many preclinical studies have shown that pectin and its derivatives have anticancer effects against leukemia, myeloma, and breast, stomach, colon, pancreatic, hepatocellular, bladder, prostate, ovarian, skin, brain, and lung cancer [77,78]. For example, pectin and pectin oligosaccharides increased the number of apoptotic cells in human colon cancer cell lines [79]. In a study using human breast cancer cell lines and mice, it was also reported that pectin induces apoptosis in vitro, suppresses cell proliferation and cell adhesion, and suppresses p53 expression in vivo, thereby reducing tumor growth and increasing the number of apoptotic cells [80]. Clinical studies to date include a phase II study reporting that the oral administration of modified citrus pectin reduced prostate-specific antigen doubling time in prostate cancer patients [81].

## 4. Hypothesis: Can the Combination of “Alkalization Therapy” and Natural Products Have a Positive Effect on Cancer Patients?

As an adjuvant therapy for cancer, would alkalization therapy and natural products have an anticancer effect, either individually or in combination? We believe that the anti-inflammatory, antioxidant, and immunostimulatory effects of alkalization therapy and natural products may complement each other and may have the potential to alleviate the health damaging side effects of current chemotherapies and contribute to improving the overall health of patients. Indeed, some researchers have expressed skepticism about the anticancer effects of alkalization therapy [82,83]. However, if alkalizing diets and the oral administration of alkalizing agents with very few side effects can provide even a small benefit, alkalization therapy may become the mainstream when conventional treatments are insufficient or when a therapeutic effect is desired at low cost.

Natural products, on the other hand, are unlikely to be used as conventional drugs owing to their complex nature and the high costs associated with their development as pharmaceuticals. However, they have the potential to support the improvement, maintenance, and recovery of health, to enhance the effectiveness of standard treatments, and to reduce their side effects. Natural products are usually taken in the form of dietary supplements and are expected to play an adjunctive role in supporting treatment. It is noteworthy that many of the natural products used as anticancer treatments are used in folk medicine, and their safety is empirically assured because they are very common in the human diet. Therefore, consuming whole plants rather than extracted, refined, and medicated versions of their constituents can be a good management method to avoid increasing the risk of health-damaging side effects in patients. Whereas we expect extracted components to have anticancer effects, the bioavailability of the raw material may be higher than that of the extracted component, as in the case of *T. yunnanensis* mentioned above. As some of the natural products mentioned here have strong anticancer effects, they may be effective adjuvant therapies that can reduce the serious side effects of chemotherapies, by enabling a reduction in their dose. Although natural products should be used carefully in combination with chemotherapy, they can be used safely and effectively with alkalization therapy, which does not have any cytotoxic effects that lead to serious health problems.

## 5. Long-Term Survival of Patients with Advanced Cancer Treated with Alkalization Therapy and Natural Products

### 5.1. Study Outline

Here, we report cases of patients with advanced cancer, who were treated with triterpenoids, parthenolides, fulvic acid, *T. yunnanensis*, or apple pectin. Data were extracted retrospectively from the medical records of patients with advanced cancer (postoperative recurrence or metastatic cancer) who visited Karasuma Wada Clinic between 1 January 2011 and 30 September 2018, were taking natural products continuously while receiving alkalization therapy in addition to standard treatment and survived for at least 5 years after starting treatment. The data includes each patient’s age, sex, type of cancer diagnosed, cancer stage, type of natural products taken, and survival time (as of 30 September 2023) since the start of treatment.

These cases were comprehensively included in the research project “Investigation of survival factors for cancer patients using data science methods” approved by the Institutional Review Board of the Japan-Multinational Trial Organization (UMIN000047446).

### 5.2. Patient Characteristics

Table 1 shows the characteristics of a cohort of 49 advanced cancer patients treated with alkalization therapy and natural products. The group includes 22 men and 27 women, with a mean age at the first clinic visit of 62.2 years (range: 39–86 years). Metastasis was observed in 31 patients, and 18 experienced a recurrence after surgery. The cancers diagnosed included non-small cell lung cancer in 18 patients, breast cancer in 9, hepatic cancer in 4, gastric cancer and colon cancer in 3 each, small cell lung cancer, pancreatic cancer, and prostate cancer in 2 each, and oropharyngeal, thymic, kidney, duodenal papillary, uterine, and unknown primary cancer in 1 patient each. Triterpenoid was the most utilized natural product, used by 48 patients, followed by parthenolide in 19, fulvic acid in 11, *T. yunnanensis* in 5, and apple pectin in 2.

### 5.3. Outcomes of Patients Treated with Alkalization Therapy and Natural Products in Combination with Standard Treatments

Figure 4 illustrates the survival duration for each patient following treatment with a combination of natural compounds and alkalization therapy, together with standard treatments. As of 30 September 2023, the average survival time across the cohort was 2886 days, ranging from 1840 to 4592 days. At the specified date, two patients (one with non-small cell lung cancer and the other with pancreatic cancer) had died from their illnesses.

Survival durations (days) of each patient who underwent treatment combining natural compounds and alkalization therapy, in addition to standard cancer treatments, are shown. The horizontal axis shows the patient number and the type of cancer, and the vertical axis shows the survival time (in days) since the start of natural products’ administration. The patient with non-small cell lung cancer (no. 1) and the patient with pancreatic cancer (no. 41), shown in orange, have died, but the other patients are alive at the time of writing.

## 6. Summary and Limitations

Here we summarized the anticancer effects of alkalization therapy and natural products, and reported cases of advanced cancer patients who were treated with alkalization therapy and natural products in addition to standard therapies and achieved long-term survival. On the basis of our clinical data, it is possible that the combination of alkalization therapy and natural products with standard therapies may have a higher anticancer effect than existing therapies on their own.

There are several limitations to the results shown in this review. First, the results were from a single-center retrospective study, and do not compare patients with and without alkalization therapy or natural products. Therefore, the possibility that patients with long-term survival owing to factors not attributable to alkalization therapy or natural products were selected cannot be excluded. Second, the natural products used by the patients in this study differ depending on the type of cancer and the case, making generalization difficult, and therefore, the results cannot be evaluated straightforwardly. Finally, it is difficult to provide any lines of evidence regarding the degree of compliance of patients undergoing alkalization therapy, and furthermore, the degree to which the body and TME actually became alkalized remains unclear, and no investigation has been conducted to date on this point.

## 7. Future Directions

We believe that in the future, alkalization therapies may need to be established that not only target the impaired regulation of the extracellular pH of the TME, but also take into account intracellular pH-associated programmed cell death (alkaliptosis). For this purpose, a standard protocol for alkalization therapy should first be developed, followed by the establishment of a method to measure the pH of the TME (and intracellular pH) more accurately, so that whether or not alkalization of the TME (or cell) has been achieved can be determined. In addition, it would be desirable to conduct more generalized, prospective comparative studies, preferably at more institutions. Furthermore, a protocol to obtain general data on natural products should be developed, and comparative studies should be designed to test each of the products, to obtain more reliable lines of clinical evidence. Further knowledge should be accumulated regarding natural products that have been reported to have anticancer activity, but have not yet been utilized in clinical settings (such as curcumin [84]), toward their clinical application, and additional studies should be performed to collect accurate data.

## 8. Conclusions

This study outlined the potential of both alkalization therapy and natural products in cancer treatment, suggesting that their combined use might yield a synergistic effect. To accurately assess their efficacy and potential as a treatment strategy for advanced cancer, further extensive research, including clinical trials, is essential.

## Figures and Tables

**Figure 1 pharmaceutics-16-00787-f001:**
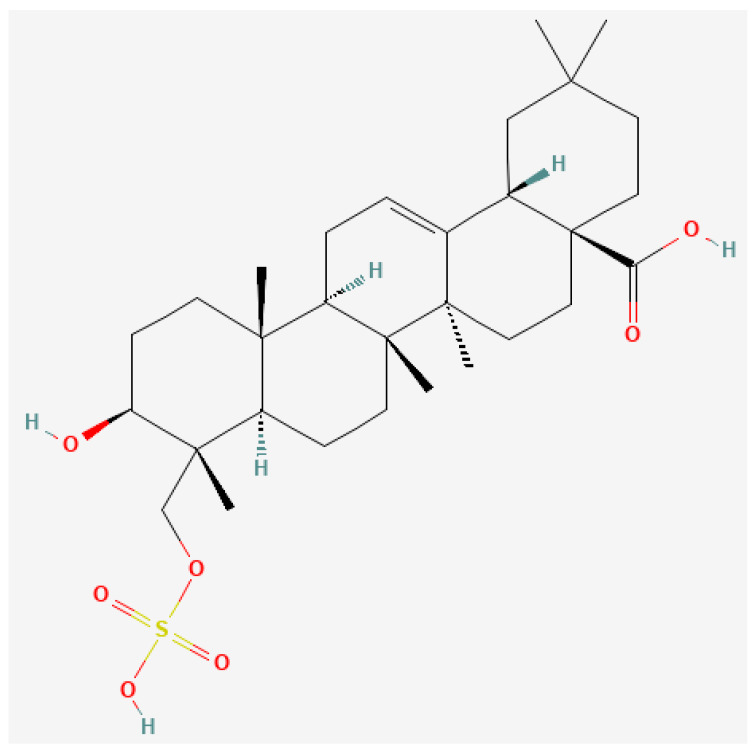
Two-dimensional structure of triterpenoid (National Center for Biotechnology Information (2024). PubChem Compound Summary for Compound ID 451674, Triterpenoid. Available online: https://pubchem.ncbi.nlm.nih.gov/compound/Triterpenoid, accessed on 6 May 2024).

**Figure 2 pharmaceutics-16-00787-f002:**
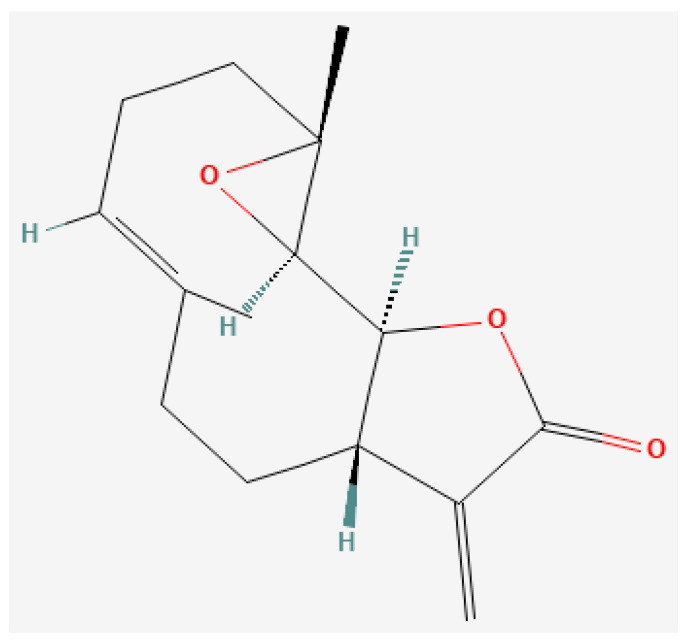
Two-dimensional structure of parthenolide (National Center for Biotechnology Information (2024). PubChem Compound Summary for Compound ID 7251185, Parthenolide. Available online: https://pubchem.ncbi.nlm.nih.gov/compound/7251185, accessed on 6 May 2024).

**Figure 3 pharmaceutics-16-00787-f003:**
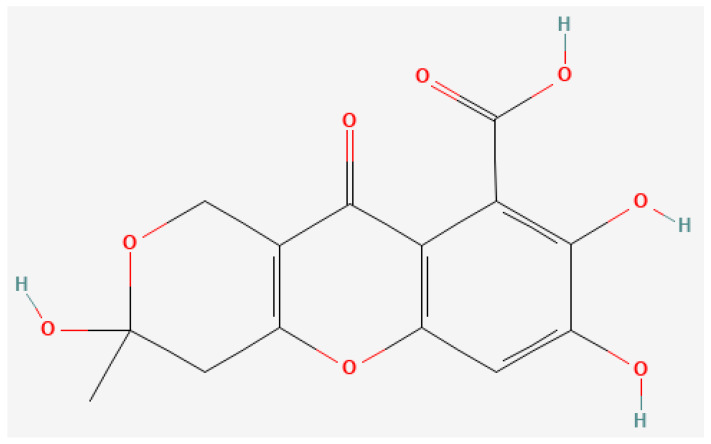
Two-dimensional structure of fulvic acid (National Center for Biotechnology Information (2024). PubChem Compound Summary for Compound ID 5359407, Fulvic acid. Available online: https://pubchem.ncbi.nlm.nih.gov/compound/5359407, accessed on 6 May 2024).

**Figure 4 pharmaceutics-16-00787-f004:**
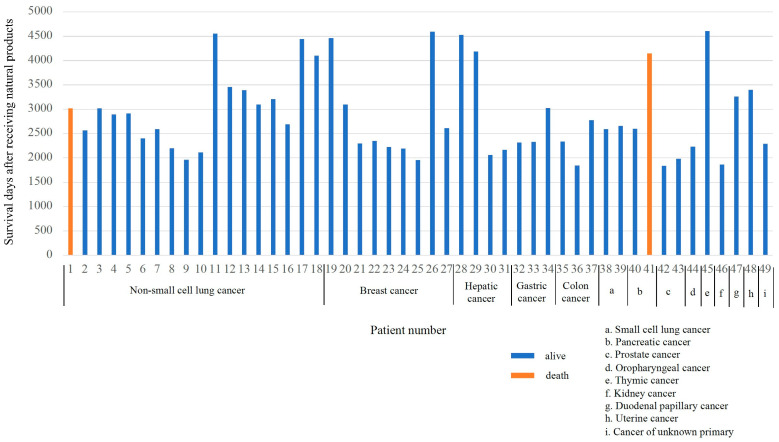
Survival duration of the patients.

**Table 1 pharmaceutics-16-00787-t001:** Patient characteristics.

No. of Patients	49
Age (range), years	62.2 (39–86)
Sex	
Men	22
Women	27
Progression of cancer	
Metastasis	31
Recurrence after surgery	18
Diagnosis of cancer	
Non-small cell lung cancer	18
Breast cancer	9
Hepatic cancer	4
Gastric cancer	3
Colon cancer	3
Small cell lung cancer	2
Pancreatic cancer	2
Prostate cancer	2
Oropharyngeal cancer	1
Thymic cancer	1
Kidney cancer	1
Duodenal papillary cancer	1
Uterine cancer	1
Cancer of unknown primary	1
Natural products	
Triterpenoid	48
Parthenolide	19
Fulvic acid	11
*Taxus yunnanensis*	5
Apple pectin	2

## Data Availability

All data are contained within the article.

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
