# Peer review of "Exploring the Potential Use of Natural Products Together with Alkalization in Cancer Therapy"

_pharmaceutics, 2024, doi:10.3390/pharmaceutics16060787_

Round 1
Reviewer 1 Report
Comments and Suggestions for Authors
I've just completed reviewing the manuscript by Hamaguchi et al., submitted for publication in Pharmaceutics. While the manuscript briefly touches on the synergistic effects of alkalization alongside natural products, the descriptions of each subsection are quite concise and require expansion. Here are some suggested modifications:
1. Authors ought to clarify whether alkalization is exclusively linked to natural products as part of the combination therapy.
2. More information regarding natural products should be provided, including details on quantities obtained from plant extraction and purification procedures, as well as their mode of action.
3. The authors should indicate whether alkalization demonstrates superior efficacy when combined with natural products or with other anti-cancer drugs.
4. An explanation is needed as to why the authors focus solely on discussing combination therapy involving alkalization.
5. It would be beneficial for the authors to discuss the availability of any clinical data or preclinical information supporting their arguments.
Author Response
Reviewer 1
I've just completed reviewing the manuscript by Hamaguchi et al., submitted for publication in Pharmaceutics. While the manuscript briefly touches on the synergistic effects of alkalization alongside natural products, the descriptions of each subsection are quite concise and require expansion. Here are some suggested modifications:
- Authors ought to clarify whether alkalization is exclusively linked to natural products as part of the combination therapy.
Response:
Thank you for your valuable comment. The primary focus of this paper is the potential effectiveness of alkalization therapy when combined with natural products, which may have the potential to become an effective and useful treatment approach for cancers. As discussed in Section 2.2, “Studies demonstrating the positive effects of alkalization therapy on cancer,” there are reports from basic research indicating that systemic alkalization alone can inhibit tumor growth, and this has also been observed in many animal experiments. Furthermore, it has been reported that the combination of alkalization therapy with chemotherapy extends the survival period of animals with advanced cancer, and a synergistic effect of alkalization therapy with chemotherapy and immunotherapy has been observed. We have included a summary of these findings at the end of Section 2 of the revised manuscript, as follows. (lines 159–164)
“Research collectively suggests that systemic alkalization may not only have anti-tumor effects and alleviate symptoms, but may also prevent tumor formation. It holds promise for boosting chemotherapy efficacy and stimulating tumor immunity. There-fore, alkalization therapy, which modifies the host’s environment, offers a strategic approach to make conditions less favorable for tumor survival, and complements other treatments, such as natural products.”
- More information regarding natural products should be provided, including details on quantities obtained from plant extraction and purification procedures, as well as their mode of action.
Response:
In accordance with this comment, we have added references and detailed information about the structure of the compounds discussed in this paper. As the exact details of the quantity of the natural compounds obtained from the extraction and purification procedures are unclear, this information could not be included, and we ask for your kind understanding on this matter. The sentences and references that were added to the revised manuscript are as follows.
(lines 234–237)
“Triterpenoids constitute compounds within the terpenoids, which is the most ex-tensive and diverse class of natural substances. They encompass a broad spectrum of structural variations, from acyclic to hexacyclic C-30 skeletons, reflecting the extensive diversity and complexity inherent in natural compounds”
Additional reference
(46) Sandeep; Ghosh, S. Chapter 12 - Triterpenoids: Structural diversity, biosynthetic pathway, and bioactivity. In Studies in Natural Products Chemistry, Atta ur, R. Ed.; Vol. 67; Elsevier, 2020; pp 411-461.
(lines 241–243)
“Ursolic acid, which is also known as 3β-3-hydroxy-urs-12-ene-28-oic-acid, is a pentacy-clic triterpenoid, which has the chemical formula C30H48O3 and a molecular mass of 456.71 g/mol”
Additional reference
(48) Kashyap, D.; Tuli, H. S.; Sharma, A. K. Ursolic acid (UA): A metabolite with promising therapeutic potential. Life Sci 2016, 146, 201-213. DOI: 10.1016/j.lfs.2016.01.017.
(lines 265-266)
“It has the chemical structure C15H20O3 and a molecular mass of 248.32 g/mol”
Additional reference
(56) Zhu, S.; Sun, P.; Bennett, S.; Charlesworth, O.; Tan, R.; Peng, X.; Gu, Q.; Kujan, O.; Xu, J. The therapeutic effect and mechanism of parthenolide in skeletal disease, cancers, and cytokine storm. Front Pharmacol 2023, 14, 1111218. DOI: 10.3389/fphar.2023.1111218.
(line 285)
“Fulvic acid (C14H12O8, molecular mass of 308.24 g/mol)”
Additional reference
(62) Dean, F. M.; Eade, R. A.; Moubasher, R. A.; Robertson, A. Fulvic Acid: Its Structure and Relationship to Citromycetin and Fusarubin.
- The authors should indicate whether alkalization demonstrates superior efficacy when combined with natural products or with other anti-cancer drugs.
Response:
As we mentioned in our response to Comment 1, alkalization therapy and chemotherapy have been clearly demonstrated to have synergistic effects in previous studies. Furthermore, the effectiveness of alkalization therapy when conducted together with treatment using natural products is a key point and a central theme of this paper.
- An explanation is needed as to why the authors focus solely on discussing combination therapy involving alkalization.
Response:
Alkalization therapy is considered to be a treatment method that can enhance the effects of chemotherapy, and activate immune responses by modifying the host’s environment. It is a therapeutic approach that can be considered for use in combination with various other treatments. Rather than being an aggressive intervention against tumors, such as anticancer chemotherapy, it is a method aimed at conditioning the host environment to make it less conducive for tumor survival. This perspective underlies our current investigation into the use of alkalization therapy in combination with natural products. As we mentioned in the response to Comment 1, the following changes were made to the manuscript to make these points clear. (lines 159–164)
“Research collectively suggests that systemic alkalization may not only have anti-tumor effects and alleviate symptoms, but may also prevent tumor formation. It holds promise for boosting chemotherapy efficacy and stimulating tumor immunity. There-fore, alkalization therapy, which modifies the host’s environment, offers a strategic approach to make conditions less favorable for tumor survival, and complements other treatments, such as natural products.”
- It would be beneficial for the authors to discuss the availability of any clinical data or preclinical information supporting their arguments.
Response:
As we described in the main text and in response to Comment 1, the concept of alkalization therapy is well-established, not only through our research but also through basic studies and animal experiments by various other research groups. Additionally, we have cited several studies other than our own research papers that have applied this therapy in clinical settings. However, research on cancer treatments combining alkalization therapy with natural products has not been conducted previously, and to our knowledge, this the first report on such a study to date.
Reviewer 2 Report
Comments and Suggestions for Authors Generally, the authors present an interesting and nice study. The paper is well written and presented. Please find my specific comments below: 1. Line 47-48, the concept of ‘TME’ was crucial important in this manuscript, however, the authors did not provide enough information about it. 2. Line 50-51, I would suggest the authors provide some specific pH values of intracellular and extracellular pH of both normal and tumor cells. 3. Line 96-97, I would suggest the authors provide the specific bicarbonate used, for example, sodium bicarbonate. 4. Line 121, the meaning of ‘berry compounds’ was not clear. 5. Line 139 and 145, what was the initial pH and final pH? 6. Line 152, can ‘citric acid’ be used as ‘alkalizing agent’?Author Response
Reviewer 2
Generally, the authors present an interesting and nice study. The paper is well written and presented. Please find my specific comments below:
- Line 47-48, the concept of ‘TME’ was crucial important in this manuscript, however, the authors did not provide enough information about it.
Response:
Thank you for your valuable comment. Regarding this point, we have inserted the following explanation of the tumor microenvironment (TME) to Section 1 of the revised manuscript. (lines 48–52)
“The TME is a dynamic and complex entity that substantially affects cancer development and progression. It is not just a collection of tumor cells, but includes immune cells, stromal cells, blood vessels, and the extracellular matrix, which interact with tumor cells, affecting tumor initiation, progression, and therapeutic responses”
- Line 50-51, I would suggest the authors provide some specific pH values of intracellular and extracellular pH of both normal and tumor cells.
Response:
As the study described compares various cells, we feel that it would be inappropriate to list the pHs of all the cell types. Therefore, we have added the most important information presented in the cited reference as follows. (line 56)
“0.3 to 0.7 pH units lower than the average extracellular pH of normal tissues”
- Line 96-97, I would suggest the authors provide the specific bicarbonate used, for example, sodium bicarbonate.
Response:
In accordance with the comment, the specific bicarbonate used was included in the revised manuscript, as follows. (line 167)
“sodium bicarbonate and/or sodium potassium citrate”
- Line 121, the meaning of ‘berry compounds’ was not clear.
Response:
We used the term “berry compounds” to denote the vitamins and flavonoids contained within berries. To make this clear, an explanation has been added to the manuscript, as follows. (line 127)
“berry compounds (vitamins and flavonoids contained in berries)”
- Line 139 and 145, what was the initial pH and final pH?
Response:
Additional details regarding the results of pHs in mouse and human studies have been added, as follows. (lines 146–148)
“bicarbonate effectively increases extra-tumoral pH (extra-tumoral pH was predicted to increase from 7.0 to 7.07 in mice; and was predicted to increase from 7.0 to 7.04 in humans”
Regarding line 145 of the original manuscript (line 154 of the revised manuscript), the cited paper did not include specific pH values.
- Line 152, can ‘citric acid’ be used as ‘alkalizing agent’?
Response:
The alkalizing agents used in our research are citric acid and baking soda. Although citric acid is a weak acid, we utilize it for its potent alkalizing action within the body. This approach is corroborated by studies introduced in the section on “Alkaline agents” of Hamaguchi et al. (2022) Front Oncol 12, 1003588, as follows.
“It has also been reported that the oral administration of sodium potassium citrate as an alkalizing agent increases concentrations in the blood and urine, leading to an increase in urine pH and neutralization of the acidic TME in a pancreatic cancer xenograft model, thereby enhancing the therapeutic effects of anticancer drugs (tegafur/gimeracil/oteracil).”
and in the study by Ando et al. (2021) Biol Pharm Bull 44, 266-270, the administration of citric acid to a mouse model of cancer was reported to lead to the alkalization of the TME, consequently resulting in an anti-tumor effect.
Reviewer 3 Report
Comments and Suggestions for Authors
1. Line 151-155: "An “alkalizing diet” is defined as a diet with a high amount of fruits and vegetables, with blue-back fish as the main source of protein, and as little meat and dairy as possible. Specifically, patients are instructed to consume 400 g of fruits and vegetables daily and to keep a dietary record for the first 4 155 weeks." Cite the reference.
2. Section 3. Natural products for the treatment of cancer: Draw the structures of natural products in Figures.
3. It would be fine if some diagrammatical presentation of mechanism of anticancer activity of natural product added in the manuscript.
4. Please the anticancer effect of curcumin, it's also a well know natural product.
5. Increase the resolution of Figure 1.
6. Add description in the Figure 1.
7. What about the ethical approval (some patients' data reported in the manuscript)?
Comments on the Quality of English LanguageEnglish is ok
Author Response
Reviewer 3
- Line 151-155: "An “alkalizing diet” is defined as a diet with a high amount of fruits and vegetables, with blue-back fish as the main source of protein, and as little meat and dairy as possible. Specifically, patients are instructed to consume 400 g of fruits and vegetables daily and to keep a dietary record for the first 4 155 weeks." Cite the reference.
Response
Thank you for your valuable comments. In accordance with your comment, the following 2 references have been added to the manuscript.
Additional references
(43) Welch, A. A.; Mulligan, A.; Bingham, S. A.; Khaw, K. T. ‘Urine pH is an indicator of dietary acid-base load, fruit and vegetables and meat intakes: results from the European Prospective Investigation into Cancer and Nutrition (EPIC)-Norfolk population study.’ Br J Nutr 2008, 99 (6), 1335-1343. DOI: 10.1017/S0007114507862350.
(44) Hamaguchi, R.; Okamoto, T.; Sato, M.; Hasegawa, M.; Wada, H. ‘Effects of an Alkaline Diet on EGFR-TKI Therapy in EGFR Mutation-positive NSCLC.’ Anticancer Res 2017, 37 (9), 5141-5145. DOI: 10.21873/anticanres.11934.
Furthermore, in the second additional reference, we have defined an alkaline diet as follows.
“An alkaline diet was defined as one that includes more vegetables and fruits and less meat and dairy products. All patients in our clinic were instructed to follow an alkaline diet as part of routine clinical care. At every visit, a doctor or nurse provided patients with guidance on an alkaline diet and assessed their adherence to it regularly.”
- Section 3. Natural products for the treatment of cancer: Draw the structures of natural products in Figures.
Response:
The structural formulas of triterpenoid, parthenolide, and fulvic acid have been added as Figures 1 to 3 of the revised manuscript.
- It would be fine if some diagrammatical presentation of mechanism of anticancer activity of natural product added in the manuscript.
Response:
The various actions of natural products remain largely unknown, and whereas the mechanisms of action of single compounds are gradually being clarified, depicting them with clarity and precision in illustrative form would be very difficult. In accordance with the comment, we have included summaries of the mechanisms of action of the anticancer effects of the natural products mentioned within the main text, as follows.
Triterpenoids (ursolic acid): (lines 246–256)
“In terms of anticancer effects, its ability to affect the activity of several intracellular enzymes enables it to modulate processes that occur within tumor cells, and activate pathways leading to apoptosis (programmed cell death), including inhibition of the MAPK/ERK and PI3K/ACT/mTOR signaling cascades. This activation of pathways that lead to apoptosis is the most important function of ursolic acid’s anticancer activity, resulting in the inhibition of pathways that lead to cancer proliferation, growth, and metastasis. Other anticancer activities of ursolic acid have also been reported, such as its effects on changes that occur upon the exposure of cells to carcinogenic chemicals (e.g., benzo(a)pyrene and substances extracted from tobacco smoke), reactive oxygen species (ROS), ionizing radiation, and Epstein-Barr virus.”
Parthenolide: (lines 269–278)
“To date, parthenolide has been shown to have high cytotoxicity and apoptosis-inducing effects against malignant tumors, such as breast cancer and chronic myeloid leukemia, and to inhibit the proliferation and self-renewal ability of cancer stem cells, suggesting its potential usefulness for the curative treatment of cancer. In addition, a study has shown that parthenolide has anticancer effects by targeting EGFR in non-small cell lung cancer. Furthermore, parthenolide inhibits the mitochondrial respiratory chain and increases the production of ROS. ROS cause DNA damage and apoptosis in cancer cells, and hence parthenolide can also be expected to have anticancer effects by reducing the expression of antioxidant enzymes and increasing the sensitivity of cancer cells to oxidative stress.”
Fulvic acid: (lines 291–295)
“Regarding anticancer effects, fulvic acid is considered to prevent the progression of cancer by inhibiting the proliferation of cancer cells and inducing apoptosis. In ad-dition, a study has suggested that the binding of fulvic acid to transferrin in human serum enables more efficient delivery of antitumor drugs to the target tumor”
Taxus plant: (lines 301–309)
“Unlike other species of the Taxus genus, T. yunnanensis has a high taxol concentration, and has unique polysaccharides that have been shown to substantially inhibit the proliferation of HeLa and HT1080 cells in a concentration-dependent manner. T. yunnanensis has a special position within the genus, as it contains a large amount of α-conidendrin, which induces apoptosis in breast cancer cell lines. There are also reports that T. yunnanensis has significantly higher oral absorption and bioavailability than pure extracted paclitaxel in rat studies, and it has been associated with the induction of tumor cell apoptosis via multiple pathways”
Apple pectin: (lines 315–322)
“For example, pectin and pectin oligosaccharides increased the number of apoptotic cells in human colon cancer cell lines. In a study using human breast cancer cell lines and mice, it was also reported that pectin induces apoptosis in vitro, suppresses cell proliferation and cell adhesion, and suppresses p53 expression in vivo, thereby reducing tumor growth and increasing the number of apoptotic cells. Clinical studies to date include a phase II study reporting that the oral administration of modified citrus pectin reduced prostate-specific antigen doubling time in prostate cancer patients.”
- Please the anticancer effect of curcumin, it's also a well know natural product.
Response:
As you pointed out, the anticancer effects of curcumin have been reported previously. However, we did not use curcumin in our present study. We consider it to be a natural product with substantial potential for use in future research, and have hence added this information to the manuscript, as follows. (lines 428–431)
“Further knowledge should be accumulated regarding natural products that have been reported to have anticancer activity but have not yet been utilized in clinical settings (such as curcumin), toward their clinical application, and additional studies should be performed to collect accurate data.”
- Increase the resolution of Figure 1.
Response:
In accordance with the comment, Figure 1 has been replaced with a figure with improved resolution.
- Add description in the Figure 1.
Response:
A detailed explanation regarding Figure 1 has been added to the manuscript, as follows. (lines 394–398)
“The horizontal axis shows the patient number and the type of cancer, and the vertical axis shows the survival time (in days) since the start of natural products administration. The patient with non-small cell lung cancer (no. 1) and the patient with pancreatic cancer (no. 41), shown in orange, have died, but the other patients are alive at the time of writing.”
- What about the ethical approval (some patients' data reported in the manuscript)?
Response:
Information of ethical approval was present in the original manuscript, as follows. (lines 366–368 and lines 443–449 of the revised manuscript)
“These cases were comprehensively included in the research project "Investigation of survival factors for cancer patients using data science methods" approved by the Institutional Review Board of the Japan-Multinational Trial Organization (UMIN000047446).”
“Institutional Review Board statement: The case series presented in this study are comprehensively included in “Investigation of survival factors for cancer patients using data science methods” approved by the Institutional Review Board of the Japan-Multinational Trial Organization (UMIN000047446).
Informed consent statement: This research was conducted using retrospective data. Patients were provided the option to opt out from the study. Therefore, written informed consent was not obtained.”
Reviewer 4 Report
Comments and Suggestions for Authors
This manuscript by Masahide Isowa and colleques represents a brief overview on the problem of alkalization therapy for a treatment of cancer. A combination of alkalization theraphy with several natural products, such as triterpenoids, parthenolide, fulvic acid and pectins from some natural products which are used in folk medicine represent a potential for novel approaches for combined therapy of cancer. The cited literature is confirmed by the medicinal statistics. Despite the authors mark a deal of scepticism of researchers about alkalization theraphy and the clinical research has to be further conducted, this approach may increase cell penetrability of a range of chemotherapeutics to make anticancer therapy more efficient and thereby deserves coverage in the present thematic issue of the journal.
The main disadvantages are listed below:
1) The authors must confirm an ethic complience during conduction of the medical experiments and indicate the necessary documentation in the corresponding chapters of this review.
2) The manuscript is poorly illustrated. For example, some pictures containing structures of natural compounds such as triterpenoids, parthenolide, fulvic acid are needed to vitalize the text of the manuscript.
3) Page 2, line 92. Anjydase?
4) Conclusions seem to be muffed and should be substantially re-written.
Author Response
Reviewer 4
This manuscript by Masahide Isowa and colleques represents a brief overview on the problem of alkalization therapy for a treatment of cancer. A combination of alkalization theraphy with several natural products, such as triterpenoids, parthenolide, fulvic acid and pectins from some natural products which are used in folk medicine represent a potential for novel approaches for combined therapy of cancer. The cited literature is confirmed by the medicinal statistics. Despite the authors mark a deal of scepticism of researchers about alkalization theraphy and the clinical research has to be further conducted, this approach may increase cell penetrability of a range of chemotherapeutics to make anticancer therapy more efficient and thereby deserves coverage in the present thematic issue of the journal.
The main disadvantages are listed below:
- The authors must confirm an ethic complience during conduction of the medical experiments and indicate the necessary documentation in the corresponding chapters of this review.
Response:
Information of ethical approval is present in the original manuscript, as follows. (lines 366–368 and lines 443–449 of the revised manuscript)
“These cases were comprehensively included in the research project "Investigation of survival factors for cancer patients using data science methods" approved by the Institutional Review Board of the Japan-Multinational Trial Organization (UMIN000047446).”
“Institutional Review Board statement: The case series presented in this study are comprehensively included in “Investigation of survival factors for cancer patients using data science methods” approved by the Institutional Review Board of the Japan-Multinational Trial Organization (UMIN000047446).
Informed consent statement: This research was conducted using retrospective data. Patients were provided the option to opt out from the study. Therefore, written informed consent was not obtained.”
- The manuscript is poorly illustrated. For example, some pictures containing structures of natural compounds such as triterpenoids, parthenolide, fulvic acid are needed to vitalize the text of the manuscript.
Response:
Structural formulas for the triterpenoids, parthenolide, and fulvic acid have been added as Figures 1 to 3 of the manuscript.
- Page 2, line 92. Anjydase?
Response:
We apologize for the error. The term was corrected to “anhydrase”. (line 96)
- Conclusions seem to be muffed and should be substantially re-written.
Response:
In accordance with the comment, the conclusion was revised for clarity, as follows. (lines 433–436)
“This study outlined the potential of both alkalization therapy and natural products in cancer treatment, suggesting that their combined use might yield a synergistic effect. To accurately assess their efficacy and potential as a treatment strategy for advanced cancer, further extensive research, including clinical trials, is essential.”
Round 2
Reviewer 1 Report
Comments and Suggestions for Authors
I am fully satisfied with revision. Now this manuscript can be accepted in its present form.
Reviewer 3 Report
Comments and Suggestions for Authors
Revised manuscript is ok now
Reviewer 4 Report
Comments and Suggestions for Authors
The authors provided all the necessary information according to the comments.
Now the manuscript can be published in Pharmaceutics